# Mitochondrial Dysfunction: A New Hallmark in Hereditable Thoracic Aortic Aneurysm Development

**DOI:** 10.3390/cells14080618

**Published:** 2025-04-21

**Authors:** Daniel Marcos-Ríos, Antonio Rochano-Ortiz, Irene San Sebastián-Jaraba, María José Fernández-Gómez, Nerea Méndez-Barbero, Jorge Oller

**Affiliations:** 1Laboratory of Vascular Pathology, Health Research Institute-Fundación Jiménez Díaz University Hospital, Universidad Autónoma de Madrid (IIS-FJD, UAM), 28040 Madrid, Spain; daniel.marcosr@estudiante.uam.es (D.M.-R.); antonio.rochano.edu@quironsalud.es (A.R.-O.); irene.sebastian@quironsalud.es (I.S.S.-J.); mariaj.fernandezg@quironsalud.es (M.J.F.-G.); 2Centro de Investigación Biomédica en Red de Enfermedades Cardiovasculares (CIBERCV), Instituto de Salud Carlos III (ISCIII), 28029 Madrid, Spain; 3Facultad de Medicina, Universidad Alfonso X el Sabio (UAX), Villanueva de la Cañada, 28691 Madrid, Spain

**Keywords:** Marfan, aneurysm, mitochondria, smooth muscle cells, vascular pathology, connective tissue

## Abstract

Thoracic aortic aneurysms (TAAs) pose a significant health burden due to their asymptomatic progression, often culminating in life-threatening aortic rupture, and due to the lack of effective pharmacological treatments. Risk factors include elevated hemodynamic stress on the ascending aorta, frequently associated with hypertension and hereditary genetic mutations. Among the hereditary causes, Marfan syndrome is the most prevalent, characterized as a connective tissue disorder driven by *FBN1* mutations that lead to life-threatening thoracic aortic ruptures. Similarly, mutations affecting the TGF-β pathway underlie Loeys–Dietz syndrome, while mutations in genes encoding extracellular or contractile apparatus proteins, such as ACTA2, are linked to non-syndromic familial TAA. Despite differences in genetic origin, these hereditary conditions share central pathophysiological features, including aortic medial degeneration, smooth muscle cell dysfunction, and extracellular remodeling, which collectively weaken the aortic wall. Recent evidence highlights mitochondrial dysfunction as a crucial contributor to aneurysm formation in Marfan syndrome. Disruption of the extracellular matrix–mitochondrial homeostasis axis exacerbates aortic wall remodeling, further promoting aneurysm development. Beyond its structural role in maintaining vascular integrity, the ECM plays a pivotal role in supporting mitochondrial function. This intricate relationship between extracellular matrix integrity and mitochondrial homeostasis reveals a novel dimension of TAA pathophysiology, extending beyond established paradigms of extracellular matrix remodeling and smooth muscle cell dysfunction. This review summarizes mitochondrial dysfunction as a potential unifying mechanism in hereditary TAA and explores how understanding mitochondrial dysfunction, in conjunction with established mechanisms of TAA pathogenesis, opens new avenues for developing targeted treatments to address these life-threatening conditions. Mitochondrial boosters could represent a new clinical opportunity for patients with hereditary TAA.

## 1. Introduction

Aneurysm derives from the Greek ανɛυρυσμα (aneurusma), meaning widening, and it can be defined as a permanent and irreversible localized dilatation of a vessel to greater than 50% of its normal size [1]. Aortic aneurysms are typically asymptomatic and undiagnosed. The progression of aortic aneurysms is associated with devastating outcomes such as aortic dissection and rupture. Aortic dissection causes sudden, severe pain and is immediately life-threatening, as it can lead to aortic rupture with extensive hemorrhage or acute loss of perfusion to various organs. Aortic ruptures account for 1–2% of deaths in Western countries [2]. Aortic aneurysms can affect virtually any segment of the aorta, including the thoracic (thoracic aortic aneurysms, TAAs) and abdominal regions (abdominal aortic aneurysms, AAAs). However, there is significant heterogeneity in the distribution of these aortic aneurysms, with the prevalence of AAAs being three times higher than that of TAAs. While both TAAs and AAAs share common pathogenic features, such as anatomic appearance, weakened aortic walls, alterations in the extracellular matrix (ECM), and dysfunction and loss of vascular smooth muscle cells (VSMCs), they are distinct disorders in terms of prevalence and etiology [3].

A systematic meta-analysis of the incidence and prevalence of TAAs in population-based studies has reported a pooled incidence of 5.3 per 100,000 individuals per year (95% confidence interval [CI]: 3.0; 8.3) and a prevalence of 0.16% (95% CI: 0.12; 0.20) [4]. Despite this relatively low prevalence, TAAs pose a substantial health risk due to their asymptomatic nature, with over 95% of cases remaining undiagnosed until acute dissection occurs—often too late for effective medical intervention. When dissection happens, it is highly fatal, causing sudden death in up to 50% of affected patients. The incidence of ruptured TAAs is estimated at 1.6 per 100,000 individuals per year (95% CI: 1.3; 2.1), with a mortality rate exceeding 90%.

Given the significant risk of rupture and dissection, surgery is recommended for all symptomatic patients and for asymptomatic patients whose aneurysm size has significantly increased. Elective surgical repair offers a favorable prognosis, with a 5-year survival rate of 85%. In contrast, patients requiring emergency surgery face a markedly poorer outcome, with a 5-year survival rate of only 37% [5].

Thoracic aortic dissections represent a considerable medical challenge worldwide, not only due to their life-threatening nature but also because of the substantial healthcare resources required for their diagnosis, monitoring, and management. Addressing thoracic aortic diseases demands significant capital investment from healthcare systems to support a relatively small proportion of the population [6].

Although treatment with β-adrenergic antagonists (β-blockers) or losartan (angiotensin-II receptor1 antagonist) might slow the progression of thoracic aortic aneurysms, the cornerstone of preventing premature deaths due to TAA dissections is aggressive surgical intervention to repair the thoracic aorta [7]. Hence, early diagnosis and timely surgical repair are critical to improving outcomes and reducing the high mortality associated with this condition. Therefore, research aimed at investigating the molecular mechanisms regulating vascular damage in TAAs would enable the development of new effective pharmacological therapies to prevent growth and fatal outcomes.

Recent studies have highlighted the critical role of mitochondrial metabolism in vascular homeostasis, underscoring its importance in maintaining cardiovascular integrity [8,9]. Specifically, mitochondrial dysfunction, coupled with metabolic rewiring, has been implicated in the development and progression of both TAAs and AAAs. These metabolic alterations appear to significantly contribute to underlying pathogenic processes, including oxidative stress and aberrant ECM remodeling [10,11,12]. Actually, novel pharmacological strategies aimed at boosting mitochondrial function and metabolic homeostasis could represent promising therapeutic approaches.

This review provides an overview of the principal mitochondrial metabolic alterations identified in hereditable TAAs, and highlights convergent pathways of VSMCs metabolic dysregulation across these genetic conditions.

## 2. Molecular Mechanisms in Hereditable Thoracic Aortic Aneurysms

TAAs can be classified as sporadic or genetic. Sporadic or acquired TAAs are primarily associated with cardiovascular risk factors such as hypertension, aging, and increased biomechanical stress on the aorta, including conditions like pregnancy [13]. In contrast, genetic TAAs occur in younger patients and can be further categorized as syndromic or familial non-syndromic. Non-syndromic TAAs account for 95% of all TAA cases [14] and are classified as either sporadic or familial, depending on whether at least one first-degree family member is affected [15]. Syndromic TAAs are linked to systemic manifestations seen in connective tissue disorders such as Marfan syndrome (MFS), Loeys–Dietz syndrome (LDS, Cutis Laxa, and vascular Ehlers–Danlos syndrome (EDS). Familial non-syndromic TAAs, also known as familial thoracic aortic aneurysms and dissections (familial TAA, FTAADs), lack systemic features and are often inherited in an autosomal dominant pattern with reduced penetrance and variable expression, affecting factors such as the age of onset, aneurysm location, and aortic diameter at dissection [16,17,18].

FTAADs were first described in families where affected members across generations exhibited thoracic aortic disease without phenotypic features of syndromes like MFS or EDS or systemic hypertension [16]. Subsequent studies revealed that up to 20% of patients with thoracic aneurysms or dissections who lack syndromic features have an affected first-degree relative, emphasizing the importance of family history. Interestingly, the distinction between syndromic and non-syndromic TAAs is often a continuum, as mutations in genes like *FBN1* (MFS) or *TGFBR1/2* (LDS) can also cause non-syndromic familial TAAs (FTAAs) [18,19].

Additional causative FTAA genes have been identified that encode proteins involved in the contraction and structural integrity of VSMCs. Among these, *ACTA2*, which encodes the smooth-muscle-specific isoform of α-actin, is the most mutated gene in FTAAs. Other causative genes include those implicated in the extracellular matrix (ECM) and TGF-β signaling pathways, such as TGFBR2, TGFB2, and SMAD3, as well as contractile apparatus components like MYH11, MYLK, and PRKG1 [16]. Despite these discoveries, over 70% of families with non-syndromic TAAs lack mutations in the currently known genes, indicating that additional causative mutations remain to be identified [19].

In addition to genetic and signaling abnormalities, different types of TAAs commonly exhibit degenerative changes in the aortic elastic media, termed cystic medial necrosis or medial degeneration. These pathological changes include elastic fiber fragmentation and disarray, massive accumulation of mucoid material, and fibrosis. Moreover, VSMCs exhibit loss of contractile capacity and dysfunction, contributing to the overall weakening of the aortic wall [20].

Marfan syndrome (MFS) is the most well known and common hereditary genetic disorder affecting connective tissue and is associated with TAAs. MFS has a prevalence of approximately 1 in 5000 individuals and does not exhibit segregation by race, ethnicity, geographic distribution, or sex [21,22]. This systemic pathology manifests a broad spectrum of clinical features, with musculoskeletal abnormalities, such as excessive bone growth, being the most recognizable. However, cardiovascular complications, particularly TAAs, represent the most severe threat to morbidity and mortality in MFS patients [21,22,23,24,25].

The paradigmatic cause of MFS is a mutation in the *FBN1* gene, located on chromosome 15. Fibrillin-1 is a large glycoprotein that assembles into microfibrils, providing structural support to tissues and regulating critical molecular pathways [21,22,26]. A key pathway influenced by Fibrillin-1 is TGF-β signaling. Latent TGF-β binding proteins (LTBPs) anchor TGF-β complexes to the ECM via Fibrillin-1, ensuring proper sequestration and controlled activation [26,27,28,29,30]. Mutations in *FBN1* disrupt this interaction, leading to dysregulated TGF-β signaling. This results in excessive activation of TGF-β, compromising ECM integrity, promoting VSMC dysfunction, and driving pathological vascular remodeling, fibrosis, and aortic dilation [31]. Interestingly, TGF-β exhibits a dual role in MFS pathophysiology. During early disease stages, heightened TGF-β activity may promote compensatory tissue repair and vascular remodeling. However, chronic overactivation contributes to maladaptive processes such as aortic wall degeneration, fibrosis, and VSMC apoptosis, as demonstrated in animal models. This biphasic role underscores the complex interplay between the protective and detrimental effects of TGF-β signaling in MFS progression [28,29,30,32].

In addition to TGF-β, angiotensin II (AngII), one of the main regulators of blood pressure, is a significant contributor to MFS-associated vascular pathology, although the precise molecular mechanism is not well understood. Elevated AngII levels, often driven by vascular ECM-remodeling and aortic dilation, exacerbate disease progression through both canonical pathways (via the angiotensin receptor type 1, ATR1) and non-canonical mechanisms, such as ROS-mediated activation of Smad-independent signaling cascades, including MAPK. This enhances non-canonical TGF-β activity and triggers a feedback loop that amplifies vascular fibrosis, ECM remodeling, and smooth muscle cell dysfunction, further accelerating aortic degeneration [33,34].

The therapeutic potential of targeting these pathways has been extensively explored in preclinical and clinical settings. Losartan, an ATR1 antagonist, has shown promising results in mitigating MFS progression. Preclinical studies in murine models of MFS demonstrated that losartan effectively reduces aortic aneurysm growth, improves ECM integrity, and normalizes TGF-β signaling [35]. Hence, losartan (ATR1 inhibitor), by its dual mechanism of action by blocking direct AngII and indirect TGF-β signaling, is a promising pharmacological strategy in MFS. Nevertheless, these findings were not supported by clinical trials in human patients with MFS, where losartan did not attenuate aortic root dilation compared to treatment with atenolol [36,37,38].

Recent research also highlights a novel facet of molecular drivers of MFS–aortic disease involved in pathological metabolic reprogramming. Emerging evidence suggests that MFS-associated mutations and ECM disruption lead to altered cellular metabolism, contributing to disease progression. Dysregulated metabolic pathways, including oxidative phosphorylation and glycolysis, have been implicated in VSMC dysfunction and aortic wall degeneration. This metabolic reprogramming represents an adaptive but ultimately pathological response to ECM stress, linking mitochondrial dysfunction and energy imbalances to the progressive aortic pathology seen in MFS [10,39]. Targeted new therapies, along with further investigation, hold promise for improving outcomes and reducing life-threatening complications in individuals with MFS.

## 3. Mitochondrial Function in Vascular Pathophysiology

Mitochondria are dynamic organelles essential for energy metabolism, signaling, and cellular homeostasis. Their primary function is ATP production via oxidative phosphorylation, a process that generates a proton gradient across the inner mitochondrial membrane to drive ATP synthase. Nevertheless, the function of mitochondria goes beyond their competence to generate molecular energy; mitochondria also regulate apoptosis and calcium metabolism and produce reactive oxygen species (ROS) as byproducts, linking mitochondrial function to vascular health [40]. Mitochondria quickly adapt to stressors by altering their morphology through fission and fusion, optimizing bioenergetic functions and regulating self-renewal through biogenesis or degradation via autophagy to maintain cellular homeostasis [41]. The metabolic shift from oxidative phosphorylation to aerobic glycolysis, analogous to the Warburg effect observed in cancer, represents a maladaptive response to mitochondrial impairment. Mitochondrial dysfunction, defined as the inability of mitochondria to adapt appropriately to cellular needs, supports many vascular disorders and drives processes such as normal and premature aging, inflammation, and apoptosis [42,43,44]. In cardiovascular diseases, particularly atherosclerosis, key pathological features include DNA damage, chronic inflammation, cellular senescence, and apoptosis [45,46,47,48]. Growing evidence highlights mitochondrial dysfunction as a pivotal contributor to vascular disease development, emphasizing its central role in maintaining normal vascular physiology and its disruption in pathological conditions [49].

### 3.1. Mitochondrial Biogenesis

Proper mitochondrial function is sustained by mitochondrial biogenesis pathways, particularly those regulated by the transcriptional coactivator PGC-1α. Rather than being produced de novo, new mitochondria are formed by adding components to existing ones [50]. Mitochondrial biogenesis relies on the coordinated expression of both mitochondrial- and nuclear-encoded proteins, as well as the replication of mitochondrial DNA (mtDNA). Mitochondria possess their own circular genome (mtDNA), which is replicated independently of the nuclear genome [51]. Peroxisome-proliferator-activated receptor gamma coactivator 1-alpha (PGC-1a) is a transcription factor which promotes the expression of nuclear genes critical for mitochondrial replication and function, including the nuclear encoded gene, transcription Factor A Mitochondrial (TFAM). TFAM expression is essential for mitochondrial DNA replication and transcription, and its depletion disrupts oxidative phosphorylation, forcing cells to base their metabolism on glycolysis [39,52]. Experimental studies using transgenic rabbits with smooth muscle Pgc-1α overexpression demonstrated an increase in the levels of mitochondrial complex proteins, reducing senescence and maintaining the contractile phenotype of VSMCs during atherosclerosis development [53]. In the context of AAAs, vascular smooth muscle cells (VSMCs) from AAA patients exhibit mitochondrial dysfunction by reducing the mtDNA content [54]. This mitochondrial dysfunction leads to a metabolic shift toward glycolysis, contributing to VSMC dedifferentiation, ECM remodeling, and medial layer thickening [55]. Similarly, in pulmonary arterial hypertension, pulmonary artery smooth muscle cells undergo metabolic rewiring, favoring glycolysis over oxidative phosphorylation.

Aberrant mtDNA packaging can activate the cytosolic cGAS–stimulator of interferon genes (STING) pathways [50]. cGAS-STING signaling is a primary driver of mtDNA stress and contributes to aneurysm progression [56]. STING activation in aortic tissue induces oxidative stress responses, cell death signals, DNA damage, and ECM degradation via MMP9 expression, making STING a critical molecule in aortic degeneration and dissection [56,57].

This shift promotes hyperproliferation, apoptosis resistance, and vascular obstruction. The overexpression of glycolytic enolase enzyme isoform *ENO1* in pulmonary hypertension further drives the synthetic dedifferentiated phenotype of pulmonary smooth muscle cells, exacerbating disease progression.

### 3.2. Mitochondrial Fusion–Fission

The functionality and the number and morphology of mitochondria play a critical role in vascular wall homeostasis. Mitochondria can change their number via biogenesis or though by fusion and fission. Fusion joins two nearby mitochondria, keeping them healthy and enhancing oxidative phosphorylation capacity. This fusion is driven in the inner membrane by optic atrophy protein 1 (Opa1) and in the outer membrane by mitofusin-1 and mitofusin-2. By other hand, fission is mediated by dynamin-related protein 1 (Drp1), mitochondrial fission 1 protein (Fis1), and mitochondrial fission factor (Mff) and helps in the redistribution of the mitochondrial content and facilitates the removal of damaged mtDNA [50,56,58].

The balance between fusion and fission is highly important in cardiovascular disease. Moreover, an exacerbation of mitochondrial fission by DRP1 activation and a reduction in mitochondrial fusion could be a contributing factor. An example is intimal hyperplasia, in which VSMCs migration can be limited by controlling mitochondrial fission, which contributes to cardiovascular disease. Regarding endothelial cells, a highly expressed circulating RNA is circTMEM165, which regulates mitochondrial fission by Drp1; finally, in inflammatory macrophages, mitochondrial fission is induced [59].

### 3.3. Mitophagy

Mitophagy, the recycling of mitochondria by autophagosomes, is another crucial process. The signal starts when mitochondrial damage occurs, such as the depolarization of the mitochondrial membrane. In this process, mitophagy is initiated to maintain mitochondrial homeostasis. In a healthy state, Parkinson’s protein 2 (Parkin) is found in an autoinhibited form in the cytosol. However, Parkin is recruited to depolarized mitochondria as a signal amplifier, while PTEN-induced putative kinase 1 (PINK1) functions as a mitochondrial damage sensor. Triggers such as reactive oxygen species (ROS) or starvation lead to PINK1 activation, which subsequently recruits Parkin. Another important component in this PINK1–Parkin-mediated mechanism is ubiquitin chains, which act as signal effectors. These ubiquitin-tagged mitochondria help autophagosomes encapsulate them for lysosomal degradation [60].

Defects in mitophagy have been linked to cardiovascular disorders. Moreover, a lack of mitophagy induces cardiovascular damage, which is linked to an increase in inflammation, mitochondrial DNA damage, and oxidative stress and a decrease in mitochondrial biogenesis. Healthy individuals exhibit lower ROS levels compared to patients with TAA, emphasizing the role of mitochondrial health in vascular integrity [57].

In an animal atherosclerosis model, aged mice, compared to young mice, showed a higher progression of atherosclerosis due to hyperlipidemia-exacerbated mitochondrial dysfunction elevating Parkin and IL-6 [61]. Pathological mechanisms of atherosclerosis are closely related to mitophagy linked to ROS, hypoxia, and glucose and lipid metabolism disorders, causing EC damage and the proliferation and phenotypic switching of VSMCs [46]. PINK1–Parkin-mediated mitophagy may control cell survival by eliminating damaged mitochondria [62,63,64]. In VSMCs, mitochondrial DNA damage may be induced by ox-LDL in atherosclerosis plaques decreasing aerobic respiration [65], VSMC phenotype conversion [66], or apoptosis [67], increasing the formation of unstable plaques [68]. VSMC phenotype and proliferation could be regulated by mitophagy preventing autophagy or apoptosis caused by low levels of ox-LDL [69,70].

In abdominal aortic aneurysm (AAA) tissue, compared to healthy aortas, Pink1–Parkin interactions are decreased. These problems in mitophagy result in ECM degradation by the release of enzymes. Cellular senescence and cell death could be mitigated by the activation of mitochondrial autophagy. Thus, the activation of Pink1 may be considered a new target for the treatment of aneurysmal disease [71]. The expression of PINK1 varies between VSMCs and macrophages, remarking distinct roles in different cell types. PINK1 may activate mitochondrial autophagy in order to eliminate abnormal mitochondrial function during the progression of AAAs. Thus, mitochondrial autophagy activation via Pink1 may reduce cell death and senescence, limiting AAA progression, resulting in cell protection [71]. PINK1 knockdown in VSMCs provokes AAA dilatation in murine models by elevating mitochondrial ROS and mitochondrial dysfunction [72].

### 3.4. Mitochondrial Alterations Associated with Reactive Oxygen Species Production

Mitochondrial dysfunction, often marked by impaired oxidative phosphorylation and excessive ROS production, plays a critical role in various vascular diseases. A crucial aspect of this dysfunction is the partial uncoupling of mitochondrial oxidative phosphorylation due to proton leakage. In this process, protons translocated by the electron transport chain bypass ATP synthase and return to the mitochondrial matrix, leading to heat generation instead of ATP production. This process increases respiration, dissipates energy, and reduces the proton gradient, contributing up to 25% of the basal metabolic rate. Proton leak also influences mitochondrial superoxide production and acts as a regulatory mechanism during oxidative stress conditions, such as diabetes, tumor drug resistance, ischemia–reperfusion injury, and aging [73]. The elevated ROS levels in human ascending TAA tissues have been linked to medial degeneration and increased expression of connective tissue growth factor. This mechanism triggers the regulation of the synthetic/proliferative phenotype of VSMCs in both human and murine TAA models induced by Ang-II infusion [74]. Evidence also suggests that autophagy regulates VSMC apoptosis in response to ROS and facilitates the transition of VSMCs to a proliferative/synthetic state [75]. Moreover, the reduction activity of glutathione peroxidase (GPx) has also been associated with TAA formation by increasing oxidative stress and the activation of TGF-β [76].

In vascular diseases, reduced mtDNA copy numbers or metabolic deficiencies are associated with enhanced ROS formation and decreased mitochondrial biogenesis. Excessive ROS production by dysfunctional mitochondria exacerbates inflammatory responses, further worsening conditions like AAA and pulmonary arterial hypertension [39,54]. Moreover, excessive ROS levels contribute to cardiac aging by causing mtDNA damage, including mutations, and a decrease in mtDNA copy number [77]. The accumulation of dysfunctional, ROS-producing mitochondria may also accelerate vascular aging in hypertension [78]. Over a lifetime, mitochondrial oxidative stress contributes to aortic stiffening by driving vascular wall remodeling, increasing the stiffness of VSMCs, and inducing apoptosis [79]. Considering the role of oxidative stress in the pathophysiology of aortic aneurysms, numerous preclinical and clinical investigations have explored the feasibility of antioxidant defenses using exogenous antioxidant compounds. The potent antioxidant properties of various vitamins have been studied in both preclinical and clinical models. Interestingly, in murine models of Marfan syndrome, a mixture of vitamins B6, B9, and B12, administered orally for 20 weeks, was found to mitigate TAA formation [80]. Overall, there remains a lack of human trials investigating antioxidant compounds with significant clinical effects on aortic aneurysms. Additionally, melatonin treatment in a BAPN-induced murine model of TAA increased aortic SIRT-1 abundance and activity, Nrf-2 levels, and SOD activity, which was associated with a lower incidence of aneurysm formation and rupture [81]. In a murine model of angiotensin-II-induced acute aortic dissection, treatment with ursodeoxycholic acid increased vascular Nrf2 expression, which was linked to a decreased expression of pro-oxidant NADPH oxidase and a reduced incidence of aortic dissection [82].

### 3.5. Mitochondrial Defects Associated with Aging

Mitochondrial changes associated with aging also contribute to diseases linked to multimorbidity in frail older patients, including vascular disorders, by impairing mitochondrial metabolism, reducing ATP synthase activity, promoting electron leakage, increasing ROS production, and diminishing overall energy efficiency [83]. Aging also decreases mitochondrial biogenesis in endothelial cells and VSMCs, impairing energy metabolism [84]. Elevated mitochondrial ROS levels are implicated in age-related vascular dysfunction, driven by a dysfunctional electron transport chain, enhanced peroxynitration [85], reduced glutathione [86], and decreased Nrf2 (Nuclear factor [erythroid-derived 2]-like 2)-mediated antioxidant defense [87]. Mitochondria-derived H_2_O_2_ contributes to vascular inflammation by activating NF-κB [88], while mtROS in VSMCs is linked to MMP activation and induces apoptosis [89]. mtDNA is highly susceptible to damage due to its proximity to ROS production, lack of histone protection, and limited repair capacity [90]. Aging increases mtDNA mutations and deletions, impairing energy production and contributing to vascular aging and atherogenesis. mtDNA deletions have been detected in human atherosclerotic lesions [91]. ApoE-deficient and transgenic for PolG (DNA polymerase subunit gamma mitochondrial), lacking in proof-reading activity, accumulate mutations in their mtDNA and exhibit accelerated atherosclerosis, associated with diminished proliferation and an increase in apoptosis of VSMCs [92]. Therapeutic strategies targeting mtROS, such as resveratrol [83] and MitoQ [93], have shown promise in improving endothelial function and cognitive performance in aging models. Sirtuins, particularly SIRT1 and SIRT3, regulate mitochondrial function, including biogenesis, ROS production, and autophagy [86], and their activity is reduced in aging. Resveratrol is an activator of sirtuin activity [94]. NAD^+^, a sirtuin cofactor, declines with age, partly due to PARP-1 overactivation [95]. Restoring NAD^+^ levels, such as through nicotinamide mononucleotide (NMN) supplementation, reverses mitochondrial dysfunction and vascular aging by activating sirtuin pathways [96]. These alterations, coupled with vascular stiffening and hypertension, impair mitochondrial function and exacerbate vascular dysfunction [40,97].

### 3.6. Mitochondrial Alterations Associated with Cytoskeleton–ECM Axis

The cytoskeleton, composed of actin filaments, microtubules, and intermediate filaments, also plays a pivotal role in mitochondrial function and derivative processes that need the energy they produce [40,98]. Actin filaments guide mitochondrial transport to regions requiring energy output and immobilize them to maintain ATP supply throughout varying conditions, adapting their activity as the requirements change. Cytoskeleton filament polymerization influences mitochondrial membrane permeability and, thus, directly affects mitochondrial metabolite input and energy output [98]. Actin polymerization is critical for maintaining cellular differentiation states; therefore, disruptions in cytoskeletal dynamics affect mitochondrial positioning and activity during processes like cell division, respiration, cell homeostasis, and apoptosis [40,52,98]. Recent research has shown that mitochondrial recruitment and cytoskeletal distribution are related to pathological conditions in cancer, allowing for cell movement, relocating mitochondria, and enhancing their metabolism towards the cytoskeleton [98]. Cytoskeletal–ECM–mitochondrial interactions thus form a critical axis for vascular health, with disturbances contributing to disease states. The ECM, composed primarily of fibrillar proteins such as collagen, fibrillin, fibronectin, and proteoglycans, provides structural and biochemical support. It also participates in mechanotransduction and molecular signaling. ECM homeostasis is crucial for maintaining vascular integrity, and disruptions can result in fibrosis and pathological remodeling [99]. Mitochondria are closely linked to ECM dynamics, responding to mechanical signals transmitted through integrins and focal adhesions that connect ECM fibers to the intracellular cytoskeleton. These signals regulate mitochondrial distribution and activity, ensuring localized ATP production in regions of high demand [40]. In fibroblasts, mitochondrial and metabolic regulators play a crucial role in ECM homeostasis. Fatty acid oxidation drives a catabolic fibroblast phenotype, promoting ECM degradation, whereas glycolysis supports an anabolic state, facilitating ECM remodeling [99]. Moreover, ECM remodeling has been reported to activate a TGF-β response, leading to mitochondrial fission and the mitochondrial unfolded protein response (UPRMT), highlighting the importance of ECM–mitochondria crosstalk, particularly in the context of aneurysms, where TGF-β plays a key role [100].

These findings highlight the potential of targeting mitochondrial pathways as therapeutic strategies for vascular diseases and underscore the complex interplay between mitochondrial function, aging, and aortic pathology. The ECM–cytoskeleton–mitochondrial axis might represent a key regulator in the pathophysiology of TAA, offering novel avenues for clinical intervention and new molecular mediators.

## 4. Mitochondrial Dysfunction in Marfan Syndrome

Recent evidence highlights alterations in mitochondrial function as a pivotal factor in the development of TAA in MFS, emphasizing mitochondrial dysfunction and metabolic remodeling as critical drivers of disease progression. Transcriptomics studies in aortas from Marfan syndrome mouse models (*Fbn1^C10341G/+^*) [10] and multi-omics approaches in human aortas from MFS patients [101,102] have consistently revealed significant reductions in key regulators of mitochondrial function, including PGC1α, TFAM, and mitochondrial complexes. Furthermore, impaired mitochondrial respiration in MFS-VSMCs triggers a metabolic shift from oxidative phosphorylation to glycolysis. This metabolic rewiring is accompanied by elevated lactate production and the activation of glycolysis-promoting pathways, such as HIF1α. Importantly, this metabolic remodeling has been associated with the pathological phenotypic switching of VSMCs, characterized by a loss of their contractile phenotype and increased ECM remodeling, senescence, and inflammation [10]. These findings identify mitochondrial dysfunction as one of the major canonical pathways disrupted in MFS.

Conditional *Tfam*-knockout mice, specifically in VSMCs, exhibit severe mitochondrial dysfunction, provide a compelling model for understanding the loss of mtDNA in aortic pathology. These mice developed aortic aneurysms and dissections, accompanied by significant histological alterations, including elastin fragmentation, proteoglycan deposition, and medial degeneration, classic features associated with MFS aortic pathology. On a cellular level, *Tfam*-deficient VSMCs undergo a pathological transformation, adopting a synthetic phenotype characterized by increased ECM production, and loss of contractile function. This phenotypic switch might be driven, in part, by the activation of the cGAS-STING pathway, triggered by mtDNA depletion and the cytoplasmic release of mitochondrial DNA fragments. This pathway induces chronic inflammation, marked by elevated levels of pro-inflammatory cytokines, and it promotes a senescent phenotype in VSMCs [19]. These cellular changes further exacerbate the structural and functional decline in the aortic wall. Remarkably, these features mirror those observed in VSMCs derived from MFS patients, reinforcing the pivotal role of mitochondrial dysfunction in TAA pathogenesis. The similarities between the Tfam-knockout model and MFS provide a strong foundation for exploring targeted therapies aimed at preserving mitochondrial integrity in TAA development.

Importantly, mitochondrial dysfunction in VSMCs is influenced by ECM composition. VSMCs cultured on ECM produced by *Fbn1*-deficient cells exhibited reduced mitochondrial respiration, lower Tfam and mtDNA levels, and increased glycolytic activity. These findings demonstrate that ECM derived from MFS-VSMCs exerts a direct effect on cellular metabolism, highlighting the intricate interplay between ECM remodeling and mitochondrial function in driving MFS pathogenesis. Notably, mitochondrial function in MFS can be effectively restored by using the NAD^+^ precursor nicotinamide riboside (NR). Treatment with NR enhances mitochondrial respiration, increases TFAM expression, and reduces glycolytic rate in both murine *Fbn1*-deficient VSMCs and fibroblasts from MFS patients. In vivo, NR administration not only reversed aortic dilation but also promoted actin polymerization within the aortic media, mitigated medial degeneration, and restored Tfam expression and mtDNA levels. Additionally, NR normalized the transcriptomic signature of genes implicated in mitochondrial function and Marfan pathogenesis in the aortas of MFS mice [10]. The negative effect that mitochondrial dysfunction has on the change of phenotype from contractile to synthetic in MFS has been corroborated in a VSMC cellular model through FBN1 silencing. Furthermore, the restoration of the contractile phenotype through mitochondrial boosting by coenzyme Q10 further supports the potential therapeutic potential of the mitochondrial–ECM axis in MFS [103].

*Cutis laxa* is a connective tissue disorder characterized by loose and wrinkled inelastic skin. *Fibulin-4^R/R^* is a cutis laxa mouse model in which the ECM protein Fibulin-4 is reduced, resulting in progressive ascending aneurysm formation and early death. In this model, VSMCs present a lower oxygen consumption and increased acidification rates, similarly to the *Tgfbr-1^M318R/+^* (Loeys–Dietz syndrome) model and fibroblasts from MFS patients. Upon gene expression analysis, the activity of PGC1α in VSMCs was downregulated, and its activation restored the mitochondrial respiration rate and improved their reduced growth potential. Here, mitochondrial dysfunction and metabolic dysregulation led to increased ROS levels and altered energy production, which resulted in aortic aneurysm formation in a similar way to the MFS model [104].

AAAs are different from TAAs in disease location, and in their etiology such as atherosclerosis and linked to risk factors such as age gender or hypertension, with an important contribution of the inflammatory content [105]. However, they share some similarities and parallel mechanisms with TAAs such as the degeneration of medial VSMCs. In AAAs, impaired mitochondrial function in VSMCs has been linked to a metabolic shift toward glycolysis. Similar to observations in MFS, VSMCs from murine ApoE-deficient mice infused with AngII, as well as VSMCs from AAA patients, exhibited reduced expression of key mitochondrial regulators, including PGC1α and TFAM, along with decreased mtDNA levels. Notably, treatment with NR, significantly reduced AAA formation and the incidence of sudden death caused by aortic ruptures in an ApoE-deficient mice model [39]. Furthermore, intervention with the glycolysis inhibitor PFK15 in an AngII model demonstrated that interference with glycolytic changes inhibited aneurysm formation, suggesting a clear parallel between metabolic changes in both human AAAs and the AngII model [106]. Moreover, the extracellular protein Galectin-1 has been identified as a critical modulator of VSMC phenotype through its influence on mitochondrial function, playing a role in both atherosclerosis and AAA pathogenesis [107]. These findings underscore the central role of mitochondrial function in VSMC behavior and demonstrate its potential as a therapeutic target for aortic aneurysms independently of their etiology.

## 5. Mitochondrial Respiratory Dysfunction in Loeys–Dietz Syndrome and Familial Thoracic Aortic Aneurysm and Dissections

Previous research has demonstrated that mitochondrial metabolism plays a key role in regulating the VSMC phenotype during aortic remodeling in MFS and AAAs, a process finely tuned by ECM composition [7,33,90]. In our latest study, published in this special issue (cells-3592876) [108], we investigated whether mitochondrial metabolism is also affected in other heritable TAA disorders.

Using in vitro models of inherited TAA diseases, we overexpressed causative mutations in primary VSMCs—human *TGFBR2^G357W^* (linked to LDS) and *ACTA2^R179H^* (associated with FTAAD). This resulted in reduced Tfam mtDNA levels and decreased mRNA expression of mitochondrial regulators. Additionally, these mutations led to impaired mitochondrial respiration, accompanied by increased extracellular lactate production, indicating a metabolic shift toward glycolysis as the primary energy source.

Notably, treatment with the mitochondrial booster NR reversed these metabolic alterations, improving contractility while reducing matrix metalloproteinase activity and other secretory markers. Thus, glycolytic metabolism and mitochondrial dysfunction play a central role in the pathogenesis of various hereditary TAAs. Mitochondrial boosters such as NR may offer a promising therapeutic strategy for treating aortic-aneurysm-related disorders, opening new avenues for targeted therapies (Figure 1).

## 6. Clinical Perspectives

Aneurysms and their complications result in over 90% of deaths among patients with TAAs. Open surgery or endovascular prothesis are still the main pathways to prevent aortic rupture, and pharmacological treatments to prevent or reverse TAAD remain elusive [109]. Surgery of TAAs follows two main strategies: open surgical repair, which has been the standard since the 1950s; or endovascular repair, which is a less invasive approach [110,111]. However, both methods present similar rates of perioperative complications and permanent paraplegia, among other drawbacks, which implies significant morbidity and mortality regardless of the method [110]. This highlines the necessity of alternative pharmacological strategies to tackle thoracic aortic aneurysms to prevent lethal aortic rupture.

As of today, pharmacological treatments are based on lowering blood pressure to reduce biomechanical stress in the aortic wall by using β-blockers or losartan. These drugs slow down the aortic disease in MFS and LDS patients, but evidence is limited in terms of efficacy in lowering root dilatation and targeting the underlying cause of progressive aortic degeneration [111,112].

Currently, aneurysms in Marfan syndrome patients are managed by a combination of β-blockers and AT2 inhibitors, and surgical repair once the aneurysm size surpasses 50 mm. However, the level of evidence regarding medical therapy is low, despite its importance in preventing aortic dilatation [113].

The largest observational study for β-blockers was carried out by Silverman et al. [114], with a study size of 417 patients. The authors claimed that the median cumulative probability of survival for patients that took β-blockers was 72 years compared to the 70 of those who had never taken them. However, the limited number of patients over 50 years in the study makes it difficult to assess a 2-year difference in life expectancy [113]. Further observational studies [115] with a more limited number of patients revealed an extreme standard deviation in the control group, implying deep heterogeneity between treated and non-treated groups [113].

Later, another possible drug emerged from a *FBN1* missense mutation mouse model. Losartan potassium arose based on its ability to inhibit TGF- β signaling and tissue fibrosis in the animal model, which diminished aortic dilatation. This led to the prescription of losartan instead of β-blockers, even before randomized clinical trials were initiated [37]. The first prospective trial (named COMPARE) assessed 233 patients and concluded that, after 3.1 years, losartan significantly reduced the rate of aortic root dilatation when compared to the usual therapy [116]. However, these results were put into question later on by the US Pediatric Heart Network, which was the largest losartan trial (608 patients between 6 months and 25 years) [117]. Echocardiography showed that there was not any significant difference between the groups. A follow-up trial with the same design confirmed no difference in aortic dilatation rate or clinical events between treatment groups [118].

Fluoroquinolones (FQs) are a widely used class of antibiotics with broad-spectrum antibacterial activity, targeting prokaryotic topoisomerases. Recent evidence suggests that FQs may contribute to the pathological development of TAAs, particularly in patients with MFS [119]. Ciprofloxacin, a commonly prescribed FQ that inhibits bacterial DNA topoisomerase, has been shown to induce both nuclear and mitochondrial damage [120]. By impairing mitochondrial topoisomerase, ciprofloxacin triggers the release of mtDNA, leading to mitochondrial dysfunction, increased reactive oxygen species (ROS) production, and cell death. Additionally, some studies suggest a link between FQs and collagen-related damage, potentially contributing to aortic aneurysm formation [121]. However, the evidence remains inconclusive. While certain studies have identified an association between FQ use and aneurysm formation, others have found no significant increase in the risk of intracranial aneurysms or dissections [122]. Considering the potential of FQs to worsen mtDNA damage and mitochondrial dysfunction, both key factors in TAA pathogenesis, their use should be approached with caution in patients with TAA or a predisposition to aortic disease, as they may accelerate aneurysm progression and elevate the risk of aortic rupture. Further research is needed to clarify the extent of these risks and guide clinical recommendations.

Recent evidence suggests that long-term doxycycline treatment may exacerbate mitochondrial dysfunction in aortic aneurysms, potentially accelerating disease progression rather than stabilizing it. Tetracyclines, including doxycycline, have been shown to impair mitochondrial function due to their inhibitory effects on mitochondrial ribosomes, leading to decreased oxidative phosphorylation and bioenergetic failure [123,124]. In a randomized trial on patients with AAA, doxycycline treatment was associated with increased aneurysm growth, prompting early study termination [125]. A systematic review and meta-analysis of randomized controlled trials [126] concluded that doxycycline is ineffective in reducing AAA growth rates. This paradoxical effect may be attributed to a dual mechanism: while doxycycline reduces inflammation—partly by suppressing metalloproteinase activity [127]—prolonged exposure could also induce VSMC exhaustion or apoptosis, particularly in cells already burdened by mitochondrial dysfunction [55,128]. Given that mitochondrial dysfunction is emerging as a key pathological hallmark in heritable TAA (cells-3592876), the use of tetracyclines in these patients warrants caution. Notably, Q-fever patients (caused by *Coxiella burnetiid* infection), who often receive long-term doxycycline therapy, exhibit high rates of vascular disease, including aneurysm rupture, potentially exacerbated by drug-induced mitochondrial toxicity [129]. These findings highlight the need for further investigation into the long-term effects of tetracyclines on mitochondrial health and vascular integrity in aneurysm patients.

Resveratrol (RES) is a dietary supplement found in some nuts and plants. When RES is administered to *Fbn1^C1041G/+^* MFS mice, aortic dilatation is reduced as effectively as when using losartan [130]. RES induces NAD-dependent deacetylase sirtuin 1 (SIRT1), which is involved in cellular metabolism in processes such as enhancing the energetic cell status, and its inhibition induces cellular senescence. The effect of RES in MFS is associated with the downregulation of detrimental aneurysm microRNA-29b (miR-29b) and improved elastin integrity and VSMC survival [111]. RES also reduced inflammation, senescence, angiogenesis, and miR-29b in murine models, which are all related to endothelial dysfunction. Endothelial dysfunction is described as an impaired vasorelaxation caused by the loss or overproduction of NO. In endothelial cells, RES promotes NO synthesis, which is a potent vasodilator synthetized by the endothelial NO synthase (eNOS) and was impaired in the MFS murine model. The inducible isoform (iNOS) is increased in the VSMC of MFS mice. Excessive NO production by iNOS produces oxidative stress and cellular damage by accumulation of peroxynitrites. RES neutralized the aortic iNOS and NO levels in *Fbn1^C1041G/+^* MFS mice [131]. Recently, a clinical trial with RES in MFS patients was performed for a 1-year period. Daily treatment showed a tendency to decrease the aortic dilation rate, but a larger randomized trial with a longer follow-up is needed to further confirm or discard the beneficial effects of RES in MFS patients [132].

Nicotinamide adenine dinucleotide (NAD^+^) and its reduced form NADH are a redox couple that are essential for a broad range of biochemical, particularly redox, reactions. NAD^+^ levels decline during ageing in multiple species in a variety of ways, and its homeostasis alterations can be found in diabetes and cancer [133,134]. NAD^+^ boosting can be a way to compensate for the mitochondrial dysfunction associated with the imbalance of this redox pair. Some NAD+-boosting strategies include the direct supplementation of NAD+ precursors [10,135], the stimulation of enzymes involved in the synthesis of NAD^+^ [136], and the prevention of the escape of intermediates from the NAD+ biosynthetic pathway [133]. The boosting of NAD^+^ through its precursors has emerged as a promising strategy to prevent and improve the conditions of patients with TAAD. Enzymes that recover NAD^+^ molecules, such as the rate-limiting enzyme on the salvage pathway Nicotine phosphoribosyltransferase (Nampt), have also shown an important role in maintaining correct mitochondrial function. Conditional deficient mice for Nampt in VSMCs have shown mild aortic dilation, more susceptibility to aortic dissection under risk factors, and early senescence and loss of function. Results inversely correlate aortic damage markers with Nampt expression in both humans and mice, which hints at similarities with MFS pathological mechanisms regarding mitochondrial dysfunction and aortic damage [136]. NAD^+^ can be produced from different forms of vitamin B, which are known as NAD^+^ precursors, such as nicotinamide (NAM), nicotinic acid (NA), and nicotinamide riboside (NR) [133]. The last one, nicotinamide riboside (NR), has been proven to restore mitochondrial metabolism and reverse aortic aneurysm in MFS mice [10]. Even though clinical results have yet to shed light on its effectiveness, it has been demonstrated that NR can rise as much as 2.7-fold with a single oral dose in humans [135]. This poses a promising future for patients with risk of TAAD, especially MFS patients, who currently do not have access to an effective pharmacological alternative to surgery. Furthermore, the potential for mitochondrial-targeted drugs has also been described in other cardiovascular pathologies, such as genetic hypertrophic cardiomyopathy. In this condition, elevated mitochondrial NAD^+^ levels or treatment with elamipretide in myectomy samples from patients have been shown to improve mitochondrial function [137].

Future research directions in TAA pharmacological approaches appear to be shifting toward a focus on mitochondrial boosters and NAD^+^ homeostasis. New data about NAD^+^-boosting strategies in other clinical contexts is rapidly accumulating, but many issues still need to be addressed. Even after the first clinical trial with NR in 2016 [135] and the multitude that followed and that are still being carried out, there is still no evidence that the promising results observed in mice can be easily replicated in humans [133]. To test the effectiveness of NAD^+^ boosters, carefully planned clinical studies with a longer duration, a higher dose, and a significantly large number of patients are needed in rare hereditary and acquired diseases [133]. Considering that there already are phase I and II clinical studies in other related diseases, this initial process could be sped up to bring the alternative treatments that the patients require (Table 1).

## 7. Discussion

Mitochondrial dysfunction and a metabolic shift toward glycolysis may represent a unifying mechanism contributing to various hereditary and acquired vascular remodeling disorders, including hereditable TAAs. Traditionally, the pathogenesis of TAAs has been associated with dysregulated signaling in the AngII and TGF-β pathways, as evidenced in both experimental models and human studies [13,138,139]. Loss of contractile phenotype, ECM accumulation and remodeling, cellular senescence, and oxidative stress have also been proposed as important contributors to disease progression [75,130,140,141]. However, the interplay between these pathways and mitochondrial function remains poorly understood and warrants further investigation.

Increasing evidence suggests a critical role for mitochondrial metabolism in maintaining VSMC homeostasis. Recent studies describe that TAAs are associated with MFS [10] and *FBLN4* mutations [104]. Here, we show that in VSMCs harboring LDS or FTAAD mutations, mitochondrial dysfunction drives metabolic reprogramming, shifting energy production from oxidative phosphorylation to glycolysis, mirroring the metabolic and mitochondrial alterations observed in MFS VSMCs. This metabolic reprogramming has profound implications for VSMC phenotype, as mitochondrial boosters like NR can reverse both the glycolytic shift and the pathological features of VSMCs in vitro and in vivo in preclinical models. These findings underscore the potential importance of mitochondrial health in preserving the differentiated contractile state of VSMCs and maintaining vascular integrity.

Notably, mitochondrial dysfunction is not limited to TAAs but has also been observed in AAAs in hyperlipidemic ApoE-deficient mice fed with a Western diet and treated with AngII. Mitochondrial impairments in VSMCs reflect those found in VSMCs extracted in AAA from patients [39], suggesting that mitochondrial dysfunction and metabolic reprogramming might be common features in diverse vascular remodeling disorders.

Given the shared involvement of AngII, TGF-β signaling, and mitochondrial dysfunction and metabolic rewiring in these conditions, it is crucial to evaluate whether these pathways intersect in the context of TAA pathogenesis. Exploring such a nexus could reveal novel insights into the mechanisms underlying mitochondrial dysfunction in TAAs and potentially uncover therapeutic opportunities.

Furthermore, the pathological VSMCs in pulmonary hypertension exhibit a glycolytic metabolic phenotype, further supporting the centrality of mitochondrial function in vascular disorders [142,143,144]. Despite these observations, the molecular pathways linking mitochondrial dysfunction to VSMC phenotypic changes remain elusive. Future studies should prioritize understanding how vascular metabolism and mitochondria influence VSMC differentiation and contractility, as this knowledge could elucidate fundamental processes driving vascular remodeling and aneurysm development.

Although promising preclinical results have been observed in mouse model trials, this has not translated all the potential. The divergence in results may be due to interspecies differences in immune, inflammatory, and healing responses [113]. Most preclinical work is based on hypomorphic *FBN1*-mutation murine models, which make the results relevant only to a subset of cases, as there are great genotypical and phenotypical variations in human MFS patients [113]. Taken together, there is a need for more robust clinical trials. The credibility of current results is set back by limitations such as phenotypical heterogeneity among patients studied and a lack of proper control groups and large-scale clinical trials [37,113]. Meta-analysis and sub-analysis of available data would solidify current results, while adequate placebo-controlled global trials with divisions by age, genetic heterogeneity, and diverging effects on blood pressure and pulse [113] could clarify the contradictory results between existing studies and the safety and adequacy of current medical treatments of Marfan syndrome.

## 8. Conclusions

The recognition of mitochondrial dysfunction as a possible common feature in TAAs, AAAs, and other vascular remodeling disorders places mitochondria at the forefront of disease research. This focus opens the door to targeted therapies, such as mitochondrial metabolism modulators, which could prevent or mitigate disease progression. Addressing mitochondrial dysfunction offers a promising strategy to improve vascular health and reduce the clinical burden of aneurysms and related conditions (Figure 2).

## Figures and Tables

**Figure 1 cells-14-00618-f001:**
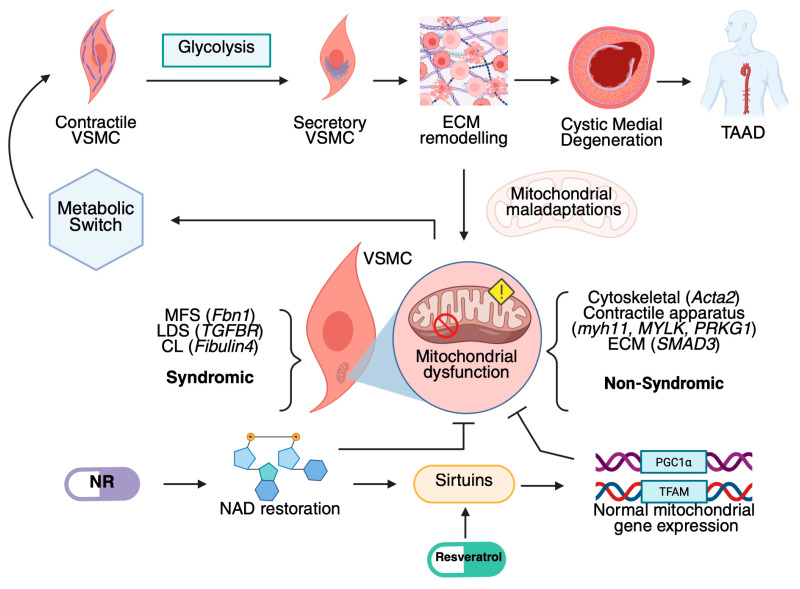
Thoracic aortic aneurysm dynamics caused by mitochondrial dysfunction [10,39,103]. NR: Nicotinamide Riboside.; NAD: Nicotinamide Adenine Riboside; Tfam: Transcription factor A mitochondrial.; MFS: Marfan Syndrome; VSMCs: Vascular Smooth muscle cells; MYLK: Myosin light chain Kinase; Acta2: Actin alpha 2; PRKG1: Protein Kinase CGMP-Dependent 1; LDS: Loews-Dietz Syndrome; CL: cutis laxa syndrome; ECM: Extracellular Matrix; MYH11: Myosin heavy chain 11 (Smooth muscle specific); SMAD3:Mothers against decapentaplegic homolog 3; Pgc1a: Peroxisome proliferator-activated receptor gamma coactivator 1-alpha.

**Figure 2 cells-14-00618-f002:**
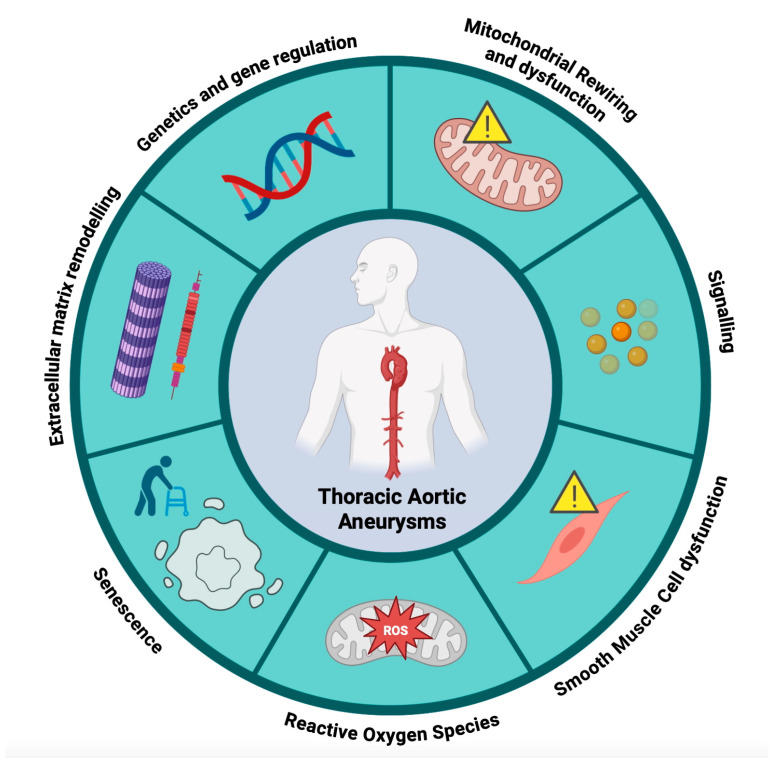
Schematic representation of the thoracic aortic aneurysm mechanism.

**Table 1 cells-14-00618-t001:** Summary of the mitochondrial pharmacological approaches, defined by target, and the effect on VSMCs.

Treatment	Target	Mitocondrial Defects Targeted	Impact on VSMCs	Preclinical Model	Human Patients/Sample	Reference
**NR (NAD+ precursor)**	NADH/NAD Imbalance SIrt1/Pgc1a/Tfam axis	Decrease Oxphos Decrease mitochondrial complexes Mitochondrial dysfunction Decrease Tfam/mtDNA Decrease Pgc1a	Improve Oxphos Decrease ECM synthesis Improve contractile phenotype Improve transcriptomic profile Decrease aortic diameter Decrease elastin breaks	*Fbn1^C1041G/+^*	Marfan patient’s fibroblasts Aortic samples from Marfan patients	[10]
**NR (NAD+ precursor)**			Improve Oxphos Decrease ECM synthesis Improve contractile phenotype	VSMCs lentiviral-transduced with ACTA2^R178H^ VSMCs lentiviral-transduced with TGFBR2^G357W^		[108]
**Coenzyme Q10**	Mitochondrial Oxphos	Decrease Oxphos Decrease Tfam/mtDNA Decrease mitochondrial mass	Improve Oxphos Decrease ECM synthesis Increase contractile phenotype		Fbn1 silencing in human VSMCs	[103]
**Forskolin**	PGC1A	Mitochondrial dysfuncion Decrease Oxphos Decrease Pgc1a Increase ROS	Improve Oxphos	Hypomorphic Fibulin-4 mice *(Fibulin-4^R/R^*) Fibulin-4 VSMCsConditional deficient mice	Marfan patient’s fibroblasts LDS patient’s fibroblasts (TRGFR2 mutant, SMAD3 mutant)	[104]
**Resveratrol** **SRT1720**	Sirtuins	Sirtuin activity decrease	Decrease aortic growth Decrease elastin breaks	*Fbn1^C1041G/+^*		[129]
	Sirt1	Aging, senescence	Increase Sirt1 nuclear staining Decrease senescence Decrease medial area Decrease apoptosis			
**Resveratrol**	Sirtuins		No impact: no significant differences in aortic diameter (1 year)		60 Adults aged 18–50 years with MFS (1 year treatment)	[131]
**Amlexanox** **TBK1 Inhibitor**	CGAS/STING pathway	Presence of mtDNA	Inflammation Aortic degeneration Aortic dissection Apoptosis Decrease ECM synthesis Increase contractile phenotype	C57/BL6J HFD + AngII *STING*-deficient (*STING^gt/gt^*) mice	Non-genetic TAA human aortic samples	[57]

## Data Availability

No new data were created or analyzed in this study. Data sharing is not applicable to this article.

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
