# Peer review of "Mitochondrial Dysfunction: A New Hallmark in Hereditable Thoracic Aortic Aneurysm Development"

_cells, 2025, doi:10.3390/cells14080618_

Round 1

Reviewer 1 Report

Comments and Suggestions for Authors

The manuscript by Marcos-Rios et al. provides a comprehensive review of the role of mitochondrial dysfunction in the pathogenesis of hereditary thoracic aortic aneurysms (TAA), with a particular focus on Marfan syndrome (MFS).  However, there are several areas where the manuscript could be improved to enhance clarity, depth, and impact.

  1. The authors mention the asymptomatic nature of TAA and the high mortality associated with aortic dissection. It would be beneficial to include some epidemiological data or statistics to emphasize the clinical significance of this condition.
  2. The authors briefly mention the role of mitochondrial dysfunction in non-syndromic TAA, but this section could be expanded. Are there specific mitochondrial genes or pathways that are commonly affected in non-syndromic TAA? How do these compare to those in syndromic TAA?

  3. The section on mitochondrial alterations associated with ROS production is informative, but it could be strengthened by including more recent studies on the role of ROS in TAA. Are there any therapeutic strategies currently being explored to target ROS in TAA?

  4. The manuscript would benefit from additional figures or tables summarizing key concepts, such as the interplay between mitochondrial dysfunction and other pathways in TAA, or a comparison of mitochondrial alterations in different forms of TAA. A table summarizing the current therapeutic strategies targeting mitochondrial dysfunction in TAA would also be useful.

  5. Page 1, Lines 10-12: While this sentence highlights the severity of TAA, it lacks specific data to support the claim. For example, epidemiological data on TAA incidence, mortality rates, or economic burden would strengthen the argument.

  6. Page 2, Lines 45-47: The "95%" statistic lacks a reference, making it difficult for readers to verify its accuracy.

  7. Page 9, Lines 384-386: This section discusses abdominal aortic aneurysm (AAA), but the focus of the paper is on TAA. The similarities and differences between AAA and TAA in terms of mitochondrial dysfunction should be clarified.
Comments on the Quality of English Language

The overall quality of the English in this manuscript is acceptable.

Reviewer 2 Report

Comments and Suggestions for Authors

This review provides a detailed and well-researched discussion on the role of mitochondrial dysfunction in hereditary thoracic aortic aneurysm (TAA). The manuscript integrates perspectives from genetics, metabolism, and vascular remodeling, offering valuable insights into disease mechanisms. 
The manuscript is classified as a review article but contains result figures, which is uncommon for a traditional review. If original data are included, the authors should explicitly indicate which figures present new research and which summarize previous studies. If new data are being reported, the manuscript may need to be reclassified as a hybrid review-research article. If all figures are sourced from prior research, proper citation and attribution should be indicated in the figure legends.
The current manuscript lacks conceptual figures, making it difficult for readers to visualize the discussed mechanisms. Schematic models illustrating mitochondrial dysfunction in VSMCs, ECM remodeling, and therapeutic interventions would significantly improve comprehension. Additionally, a graphical summary of potential mitochondrial-targeted therapies would enhance the practical relevance of the discussion.
The manuscript touches on therapeutic strategies but lacks a structured discussion on how these findings translate to clinical applications. Expanding on how mitochondrial-targeted therapies, such as NAD+ boosters and sirtuin activators, could be integrated into current TAA management would provide more practical insights. Additionally, discussing challenges in clinical translation, such as the lack of large-scale clinical trials, patient selection criteria, and potential safety concerns, would make the review more clinically relevant.
The manuscript includes a substantial selection of references, but some older citations (pre-2015) should be replaced with more recent literature. Ensuring that the latest research (2023–2024) on mitochondrial dysfunction, vascular remodeling, and pharmacological interventions is included would strengthen the scientific foundation of the review.

Comments on the Quality of English Language

"Although treatment with β-adrenergic antagonists or angiotensin receptor blockers might slow TAA progression, surgical intervention remains the primary strategy to prevent premature deaths, emphasizing the necessity of early diagnosis and new pharmacological approaches."

Suggested Revision:
"While β-blockers and angiotensin receptor blockers may slow TAA progression, surgical intervention remains the primary treatment. This highlights the urgent need for early diagnosis and new pharmacological therapies."

"Marfan syndrome is a heritable disorder caused by mutations in the FBN1 gene, which encodes fibrillin-1, a crucial component of microfibrils that provides structural support to connective tissues and is essential for extracellular matrix integrity."

Suggested Revision:
"Marfan syndrome is a genetic disorder caused by FBN1 mutations, affecting fibrillin-1, a key protein for connective tissue structure and extracellular matrix integrity."

"Losartan has shown benefits in preclinical models of MFS by reducing aneurysm progression. However, clinical trials in MFS patients did not confirm these findings. Mitochondrial dysfunction is emerging as another key factor in TAA pathogenesis."

Suggested Revision (Improved Flow):
"Losartan has shown promising effects in preclinical MFS models by reducing aneurysm progression, but clinical trials have failed to confirm its effectiveness. Given these limitations, researchers investigate mitochondrial dysfunction as a key factor in TAA pathogenesis."

Round 2

Reviewer 1 Report

Comments and Suggestions for Authors

The author has made the requested changes.

Reviewer 2 Report

Comments and Suggestions for Authors

The revised manuscript presents a comprehensive and well-structured review of mitochondrial dysfunction in hereditary thoracic aortic aneurysm (TAA), integrating insights from vascular biology, genetics, metabolism, and translational medicine. The revisions have substantially improved the work's clarity, depth, and scientific value. Previous concerns regarding the inclusion of original data have been resolved. All figures are now clearly conceptual, and the figure legends distinguish between original illustrations and adapted content where appropriate.
The addition of high-quality schematic figures greatly enhances the manuscript. These illustrations effectively summarize key mechanisms—such as mitochondrial regulation of vascular smooth muscle cell phenotype and extracellular matrix remodeling—making the complex pathways more accessible to readers.
The discussion of clinical relevance has been significantly expanded. Therapeutic strategies targeting mitochondrial function, such as NAD⁺ precursors, sirtuin modulators, and antioxidants, are clearly outlined. Table 1 provides a well-organized summary of pharmacological agents and their mechanisms. The manuscript also acknowledges critical translational challenges, including patient selection and the lack of large-scale clinical trials.
Recent literature has been thoroughly incorporated, including studies from 2023 and 2024, ensuring that the review reflects the latest developments in the field.
Overall, this review provides a timely and informative synthesis of the current understanding of mitochondrial biology related to hereditary TAA.